# Adaptation and High Yield Performance of Honglian Type Hybrid Rice in Pakistan with Desirable Agricultural Traits

**Muhammad Ashfaq** [1,*]**, Renshan Zhu** [2]**, Muhammad Ali** [1]**, Zhiyong Xu** [3]**, Abdul Rasheed** [1]**, Muhammad Jamil** [1]**, Adnan Shakir** [1] **and Xianting Wu** [2,*]

1 Faculty of Agricultural Sciences, University of the Punjab, Lahore 54590, Pakistan
2 Department of Genetics, College of Life Sciences, Wuhan University, Wuhan 430072, China
3 Wuhan Yansheng Agriculture Technology Ltd., Wuhan 430076, China
* Correspondence: ashfaq.iags@pu.edu.pk (M.A.); xiantwu@whu.edu.cn (X.W.)

**Abstract:** Honglian type cytoplasmic male sterility (CMS) is one of the three known major CMS types of rice (*Oryza sativa* L.) commercially used in hybrid rice seed production. Hybrid rice generated by the Honglian type CMS is a special group of hybrid rice, having distinct agricultural characteristics. The main objective of the study was to screen out the Honglian hybrid rice adapted for growing in Pakistan based on desirable traits. Different Honglian-type hybrid rice varietieswere tested locally in different locations in Pakistan based on various desirabletraits. Three Honglian types of hybrids (HP1, HP2, HP3) performed well, had better agricultural traits and showed high yield potential over the check variety. Different qualitative and quantitative traits were studied to conclude the advantages of these varieties for Pakistani local adaptation evaluations. Forty-eight SSR markers were used to study the genetic diversities of the hybrids. Nine selected polymorphic SSR markers (RM-219, RM-236, RM-274, RM-253, RM-424, RM-567, RM-258, RM-481, RM-493) showed genetic variations among Honglian hybrid rice varieties through PCR analysis. In 2019 and 2020, the increment of the yield potential of HP1, HP2 and HP3 was better (+43.90%, +35.44%, +37.13% and +30.91%, +33.37%, +33.62%, respectively, in both years)than the check variety KSK-133. All the desirable traits were analyzed through Principal Component Analysis (PCA). The principal components with more than one eigenvalue showed more variability. The average variability of 74.78% was observed among genotypes and their desirable traits in both years. National Uniform Yield Trial (NUYT) and Distinctness, Uniformity, Stability (DUS) trials are being conducted under the supervision of National Coordinated Rice (NCR) and Federal Seed Certification and Registration Department (FSCRD), Government of Pakistan. In the 2020 trial, the average yield of 104 rice varieties/hybrids was 8608 kg/ha; HP1, HP2 and HP3 (8709 kg/ha, 8833 kg/ha, and 9338 kg/ha, respectively) were all higher than the average yield, and HP3 yield was higher than over check varieties (D-121, Guard-53). In the 2021 trial, the average yield of 137 varieties was 7616 kg/ha; the HP1 yield (7863 kg/ha) was higher than the average overcheck varieties/hybrids. Various qualitative and quantitative traits showed desirable genetic diversity among the rice hybrids. It was also observed that, under higher temperatures, the seeds setting rate of Honglian-type hybrid rice was stable, which is the guarantee for stable yield and rice production in Pakistan. Moreover, it was considerably better, suggesting that Honglian-type hybrid rice varieties can be grown in Pakistan because they are less risky under climate change, especially the global warming challenges.

**Keywords:** rice; hybrid; desirable; agricultural traits; adaptation; Honglian

## 1. Introduction

Rice (*Oryza sativa* L) is an important food grain crop of the World population. Low yield and other undesirable traits are the common problems of rice crops that cause huge yield losses every year. Therefore, yield loss and undesirable agricultural traits cause increasing problems in agricultural practices, and selection for high-yielding rice varieties

with good quality becomes an urgent challenge for plant breeding programs. Rice is the leading staple food for both China and Pakistan, and more than half of the world's population depends on the rice crop [1,2]. It is grown under various agro-ecological conditions in different subtropical and tropical countries, including Pakistan and India. To fulfill the future food demand of the ever-increasing world population there is an urgent need to take necessary steps to increase this crop's productivity [3]. Crop improvement programs also depend on the use of germplasm resources available in various parts of the world. The improvement and expansion of world supply will also depend on developing and improving rice varieties with higher yield potential and on several conventional and biotechnological methods for developing high-yielding varieties having resistance against biotic and abiotic stresses [4].

Honglian Type hybrid rice belongs to the three-line hybrid rice category, which is internationally recognized as one of the three cytoplasmic male sterility types of hybrid rice, with "Wild Abortive Type" by Academician Prof. Yuan Longping and "Baotai Type" by the Japanese group. The Baotai type is not used in indica hybrid rice, while the wild abortive type is only used in indica rice because of its sporophyte sterile type [5–7]. The honglian type belongs to the gametophyte sterile type, which can be used in indica and japonica rice [8]. In the agricultural field trial, when a single cytoplasmic type is popularized in a large area, the homogenization of the matched varieties may lead to weak resistance to some diseases and cause severe loss or crop failures. The Honglian type is another crucial genetic resource besides wild abortive type in China. Developing Honglian-type hybrid rice can enrich the genetic diversity of rice varieties and reduce the potential threats to agricultural practices caused by single cytoplasm type growth, which is a significant guarantee of the food security for both China and the world!

Hybrid rice technology has been developed and utilized in China for more than half a century, and it is one of the main technologies used to feed population and secure the country's food safety. Hybrid rice technology has a substantial yield and enhances the yield production by 25% more than inbred rice varieties [9].

The success of the hybrids depends upon a combination of desirable traits, high yield, adaptability, stability, distinctness, uniformity, novelty and various allelic interactions [10,11]. High yield and desirable traits are essential for crop breeding and enhance agricultural production under stressful environments [12–16]. In the current scenario food, security and sustainable agricultural productions are very crucial for peoples of all the nations. However, world food security is threatened by multiple factors, i.e., increasing population, high temperature, heat stress, climate change, loss of arable lands, urbanization, lack of acclimatization and increasing demands of food and feed. Rice meets the requirements of 21% of the total calorie intake of the world population and up to 76% of that of Southeast Asia [17,18]. Enhancement of rice production would have a great impact and would be very helpful in world food security. This will only become possible by the introduction of new high yielding varieties with good agricultural traits, high adaptability, high fertility rate under stress environment and approaches to produce them.Rice genetics and genomics have been advancing over the last decade and have an important role for the development of new varieties since the determination of rice genome sequence [19,20]. However, the increase of yield per hectare of rice in China and other Southeast Asian countries is too slow to meet future demands. Genetic diversity, the Honglian type of hybrid rice and different techniques are available to achieve the desired outcome.

The rice suffers from extensive stress to sustain full yield potential due to biotic and abiotic factors [21], the main constraints of the yield in South Asia and other countries cause severe loss. Therefore, it is advantageous to select those cultivars which can tolerate multiple stresses at the same time. Different strategies will be fruitful in managing the abiotic stresses in such environments. For the last few decades, rice breeding programs and other techniques of rice provided knowledge and resources to understand the various problems, including overcoming the problem of hybrid sterility; the discovery of cytoplasmic male sterility system and restorer genes for the production of new Honglian type of hybrid rice;

the increase of yield with the development of mapping populations; and the exploitation of heterosis, genetic resources and identification of new QTLs.

On the other hand, breaking undesirable linkages between abiotic stress and plant height, abiotic stress and earliness, abiotic stress and low yield and other quality traits could also be helpful for the development of resistant varieties [22,23].

The main characteristics are prominent genetic diversity, broad hybridization selectivity and excellent comprehensive agronomic traits. Honglian-type hybrid rice can be used as a mid-season rice in the Yangtze River basin of China, and an early-season rice and late-season rice in south China. Due to its heat tolerance characteristic, it is also suitable for spreading in tropical or subtropical regions along "the Belt and Road" countries such as in South Asian and African countries. In the past two decades, old varieties generated from the second-generation sterile lines of Honglian Type CMS have been widely spread and welcomed by these countries, and their growth area was enlarged every year. Now, new Honglian type varieties with better quality, higher yield and wider adaptabilities have been selected from the third-generation to fifth-generation sterile lines and developed to carry out adaptation tests in Pakistan. These breakthrough varieties of Honglian Type hybrid rice are generated based on the solid embodiment of the scientific concepts, aiming at combinations between "high yield, high quality, wide adaptability and ecological balance", which are the leading concepts for guiding the direction of hybrid rice breeding. The main objective of the study was to screen the Honglian type of hybrid rice on the basis of adaptability, genetic diversity and high yield performance in different locations of Pakistan.

## 2. Materials and Methods

### 2.1. Plant Materials

A set of 10 hybrids, including HP1, HP2, HP3, HP4, HP5, HP6, HLR-006, WR-1906, Guard53 (check variety), D-121(check variety) and one open-pollinated variety, KSK-133 (originated from Wuhan University China, Yuan Longping High-Tech Agriculture Co., LTD China WINALL Hi-Tech Seed Co., LTD China and Rice Research Institute, Kala Shah Kaku, Pakistan), were tested in the year 2019-20 in different locations (Gujranwala, Lahore, Pakpattan) of Pakistan under Randomized Complete Block Design (RCBD). Different qualitative and quantitative parameters and seed morphological characters were studied to see the diversity among rice lines. In the year 2021 some of these hybrids were tested in National Uniform Yield Trials (NUYT) at various locations in Pakistan, with sowing date (23 June 2020), transplanting date (18 July 2020) and harvesting date (28 October 2020). In the first year of the trials, seven different sites were selected for testing the hybrid varieties, and in the second year ten different sites were selected for testing the hybrids (See Supplementary Tables for details).

### 2.2. Traits Measurement

Various agro morphological traits (seed length, width, thickness, length–width ratio, curling%, bursting%, cooked grain length, brown rice, milled rice, head rice and yield/ha) were measured with the help of a meter rod, scale and weighing balance at physiological maturity of each rice hybrid/line. The seed length–width ratio was measured with the help of the following equation in millimeters. These traits showed significant differences among the rice genotypes based on their origin and genetic diversity.

$$\text{Seed Length Width Ratio} = \frac{\text{Seed Length (mm)}}{\text{Seed Width (mm)}}$$

### 2.3. Seed Morphological Traits

Various seed morphological traits (seed length, seed width, seed thickness, seed length–width ratio and 1000 grain weight) were measured with the help of a Digital Vernier Caliper (Jinhua Longtai Tools Co., LTD. Zhejiang, China, IP-67) and weighing balance (Locosc Ningbo Precision Technology Co., LTD. Ningbo, Zhejiang, China, LP-7610). Three lots of

ten seeds data of each genotype were selected randomly. The seed morphology of each genotype is shown in Figure 1. The humidity of the rice paddy was measured with the help of a grain moisture meter (Model number: FG-506, Kett Japanese Company).

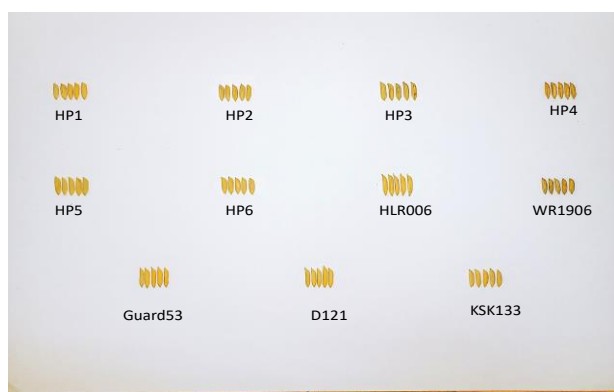

**Figure 1.** Seed morphology of ten rice hybrids and one open pollinated variety.

### 2.4. Principal Component Analysis, Variance and Correlation

Principal Component Analysis (PCA), variability and correlation of the rice hybrids were studied based on measured traits. On the basis of all the measured traits, three hybrids were selected for further genetic analysis in different sites in Pakistan to check the stability and adaptability in changing climatic conditions.PCA is a technique used for large datasets, presenting more variability among the genotypes and traits and also minimizing information loss. Desirable traits of all the hybrids were also analyzed by using Principal Component Analysis (PCA) to determine the genetic variability for these traits. Mean values of all the recorded traits of the hybrids/genotypes were used for PCA analysis.

### 2.5. NUYT and DUS Trials

Based on in-house trial results, three hybrids, HP1, HP2 and HP3, were selected for further analysis in NUYT (National Uniform Yield Trials) and DUS (Distinctness, Uniformity and Stability) trials in the year 2020-21, along with other hybrids and check varieties in different locations, i.e., the Soil Salinity Research Institute (SSRI PindiBhatian), Bahawalnagar, the Rice Research Institute Kala Shah Kaku (RRI KSK), the Rice Research Institute, Kala Shah Kaku (RRI KSK Sialkot), the Rice Research Institute (RRI Dokri), the Pakistan Agricultural Research Council (PARC KSK), the National Institute for Biotechnology and Genetic Engineering (NIBJE), the Nuclear Institute for Agriculture and Biology, (NIAB), Agriculture Research Institute (ARI Usta Muhammad), the Agriculture Research Institute, Dera Ismail Khan (ARI DI Khan), the Guard Rice Golarchi, Sun Crop, Meskay and Femtee Trading Company (FMTC) Shikarpur, the Emkay Farooqabad, Dokri Larkana, Soil Salinity Research Institute (SRRI Thatha) and Tara Crops of Pakistan based on yield and quality parameters under Randomized Complete Block Design (RCBD) with three replications. A total of 104 hybrids in 2020 and 137 hybrids in 2021 (including Honglian hybrid rice and check varieties) were tested under different ecological zones in Pakistan.

### 2.6. DNA Extraction and Quality Analysis

The leaf samples of the three selected hybrids (HP1, HP2 and HP3) were collected and stored at 4 °C. All the samples were collected at tillering stage. A CTAB method [24] was used to extract DNA. A nanodrop spectrophotometer (ND-1000) was used to check the quality and quantity of DNA at 260 and 280 nm. In all samples, DNA with good quality (concentration higher than 100 ng/µL) was used in PCR analysis.

### 2.7. DNA Fingerprinting and PCR Analysis

A set of 48 primers was used for PCR analysis [25] that covered almost the whole rice genome. Among them, nine primers were highly polymorphic, showing vast genetic

differences in the tested rice hybrids. These polymorphic primers showed the highest genetic diversity among the rice hybrids (Table 1). A total of 100 ng DNA of each hybrid was used in the experiment. The PCR amplification reaction was set as heating 94 °C for 4 min, followed by 30 cycles of denaturation at 94 °C for 1 min, annealing at 55 °C for 1 min, extension at 72 °C for 1 min and final extension at 72 °C for 10 min. The 6% gel was used in this experiment, to see more clear bands of each hybrid. Furthermore, PCR samples were scanned on a gel documentation system to see the genetic differences among the rice hybrids. These SSR primers were used by genotyping and identifying rice hybrids/varieties using the SSR marker method [26] (NY/T 1433-2014).

**Table 1.** SSR markers showed variability among rice hybrids based on various genotypic traits.

| S. No. | Primer Name | Chromosomal Location | Annealing Temp °C | Primer Sequence (5′–3′) | Fluorescence | Product Size | Nature of Polymorphism |
|---|---|---|---|---|---|---|---|
| 1 | RM-219 | 9 | 55 | F:cgtcggatgatgtaaagcct R:catatcggcattcgcctg | FAM | 194–215 | Polymorphic |
| 2 | RM-236 | 7 | 55 | F:cttacagagaaacggcatcg R:gctggtttgtttcaggttcg | VIC | 151–166 | Polymorphic |
| 3 | RM-274 | 5 | 55 | F:cctcgcttatgagagcttcg R:cttctccatcactcccatgg | V1C | 149–162 | Polymorphic |
| 4 | RM-253 | 6 | 55 | F:tccttcaagagtgcaaaacc R:gcattgtcatgtcgaagcc | PET | 133–142 | Polymorphic |
| 5 | RM-424 | 2 | 55 | F:tttgtggctcaccagttgag R:tggcgcattcatgtcatc | NED | 240–280 | Polymorphic |
| 6 | RM-567 | 4 | 55 | F:atcagggaaatcctgaaggg R:ggaaggagcaatcaccactg | PET | 248–260 | Polymorphic |
| 7 | RM-258 | 10 | 55 | F:tgctgtatgtagctcgcacc R:tggcctttaaagctgtcgc | FAM | 128–146 | Polymorphic |
| 8 | RM-481 | 7 | 55 | F:tagctagccgattgaatggc R: ctccacctcctatgttgttg | FAM | 146–165 | Polymorphic |
| 9 | RM-493 | 1 | 55 | F: tagctccaacaggatcgacc R:gtacgtaaacgcggaaggtg | VIC | 210–264 | Polymorphic |

*2.8. Statistical Analysis*

The Principal Component Analysis (PCA) and correlation were analyzed by using SAS (Statistical Analysis System) version 9.2 [27] to see the genetic variability in rice hybrids. The average data of all the hybrids were calculated based on the mean values. Statistical significance is the determination of relationships between two or more variables for the prevalence of significant variance for all studied traits that implicates the usefulness of the rice hybrids for genetics analysis at level of significance of 1% and 5%.

**3. Results**

*3.1. In-House Yield Trials*

In-house yield performance data of the rice hybrids HP1, HP2 and HP3 were developed by the University of Punjabat different locations (Gujranwala, Lahore, Pakpattan) in Punjab, Pakistan. All the parameters were taken at the time of maturity. In all the locations, the yield was measured in tons/ha. All the varieties performed very well over check variety in all the locations (Table 2).

**Table 2.** Yield comparison of hybrid rice varieties with check variety.

| Sr. No | Variety Name | Origin | Average Yield tons/ha 2019 | Average Yield tons/ha 2020 | % Increase/Decrease With Check Variety 2019 | % Increase/Decrease With Check Variety 2020 |
|--------|--------------|--------|----------------------------|----------------------------|---------------------------------------------|---------------------------------------------|
| 1 | HP1 | China | 12.75 | 10.63 | +43.90% | +30.91% |
| 2 | HP2 | China | 12 | 10.83 | +35.44% | +33.37% |
| 3 | HP3 | China | 12.15 | 10.85 | +37.13% | +33.62% |
| 4 | HP4 | China | 9.98 | 10.10 | +12.64% | +24.38% |
| 5 | HP5 | China | 10.6 | 9.85 | +19.63% | +21.30% |
| 6 | HP6 | China | 10.22 | 10.5 | +15.34 | +29.31% |
| 7 | HLR006 | China | 8.14 | 8.10 | −8.12% | −0.24% |
| 8 | WR1906 | China | 9.81 | 10.2 | +10.72 | +25.61% |
| 9 | Guard53 | China | 9.91 | 10.25 | +11.85% | +26.23% |
| 10 | D121 | China | 10.40 | 10.35 | +17.38 | +27.46% |
| 11 | KSK133 | Pakistan | 8.86 | 8.12 | - | - |

*3.2. Genetic Diversity Study of HonglianType Hybrid Rice*

Forty-eight SSR markers were used to measure the genetic diversity of Honglian type rice. The varieties showed maximum genetic diversity concerning their specific markers, product size, polymorphism and chromosomal locations. Finally, nine polymorphic SSR markers were selected to determine the genetic diversity of Honglian type rice HP1, HP2, HP3 that showed the most remarkable genetic diversity. The RM-236 and RM-424 were found to be more appropriate for HP1, HP2 and HP3 that showed more diversity among them, and very useful for traits variation (Supplementary Table S3).

*3.3. Principal Component Analysis (PCA) with Respect to Yield and Other Traits*

Principal component analysis was used to determine the phenotypic diversity under adaptability trials 2020–2021. All the hybrids showed variability according to their quality and yield contributing traits. The Honglian hybrid rice varieties (HP1, HP2, HP3) performed excellently among the full hybrids and over-check varieties under different ecological zones, having significant differences among them. The quality traits, i.e., seed length, width, thickness, length–width ratio, stickiness, curling %, bursting %, cooked grain length, brown rice, milled rice, head rice and yield per hectare of HP1 and HP3, were higher over the check varieties (D-121, Guard-53). The results are shown in Table 3. The average seed length was 7.02 mm, the cooked grain length 11.01 mm and other quality traits were higher over check varieties in both years. The yield per hectare of HPI (7863 kg/ha) was higher over check varieties in 2021 (Supplementary Table S2).

**Table 3.** Yield and quality traits comparison of hybrids with check varieties in adaptability trials 2020-21.

| | | | | | | | | | | | | | |
|---|---|---|---|---|---|---|---|---|---|---|---|---|---|
| **Adaptability Trials in the Year 2020** | | | | | | | | | | | | | |
| Sr. No | Variety | Length (mm) | Width (mm) | L/W Ratio (mm) | Thickness (mm) | Stickiness | Curling % | Bursting % | C.G.L. (mm) | Brown Rice (gm) | Milled Rice (gm) | Head Rice Recovery % | Yield Kg/hac |
| 1 | HP1 | 7.01 | 2.04 | 3.44 | 1.82 | sticky | 2 | 6 | 10.2 | 81 | 77.3 | 53.8 | 8709 |
| 2 | HP2 | 6.73 | 2.06 | 3.27 | 1.74 | sticky | 5 | 16 | 9.8 | 80 | 75 | 53.3 | 8833 |
| 3 | HP3 | 6.8 | 2.11 | 3.22 | 1.76 | sticky | 2 | 4 | 10.3 | 84.4 | 79.4 | 61.2 | 9338 |
| 4 | D-121 | 6.72 | 1.98 | 3.4 | 1.75 | sticky | 4 | 7 | 10.8 | 80 | 73.3 | 56.4 | 9171 |
| 5 | Guard-53 | 6.9 | 2.08 | 3.32 | 1.77 | sticky | 3 | 8 | 10.3 | 81.6 | 74.5 | 62.3 | 8395 |
| **Adaptability trials in the year 2021** | | | | | | | | | | | | | |
| Sr. No | Variety | Length (mm) | Width (mm) | L/W ratio (mm) | Thickness (mm) | Stickiness | Curling % | Bursting % | C.G.L. (mm) | Brown rice (gm) | Milled rice (gm) | Head rice recovery % | Yield Kg/hac |
| 1 | HP1 | 7.3 | 2.13 | 3.43 | 1.81 | sticky | 4 | 12 | 12.9 | 82.03 | 76.6 | 51.67 | 7863 |
| 2 | HP2 | 6.94 | 2.09 | 3.32 | 1.76 | sticky | 5 | 4 | 11.1 | 81 | 74.56 | 60 | 7288 |
| 3 | HP3 | 6.6 | 1.75 | 3.77 | 1.78 | sticky | 25 | 11 | 10.2 | 80.13 | 74.66 | 64.67 | 7387 |
| 4 | D-121 | 6.96 | 2.01 | 3.46 | 1.78 | sticky | 4 | 12 | 11.5 | 81.63 | 74.86 | 51 | 7518 |
| 5 | Guard-53 (1) | 6.66 | 2.02 | 3.30 | 1.77 | sticky | 8 | 4 | 9.7 | 82.86 | 76.46 | 65.30 | 7341 |

The yield per hectare (9338 kg/ha) was measured as higher over control. Based on the yield performance, the variety was positioned at number four in adaptability trials among 104 hybrids (Supplementary Table S1). Similarly, HP3 performance was higher over check varieties in the year 2020.

In the year 2020, adaptability trials of 102 hybrids and two check hybrids were conducted in different locations, i.e., the Rice Research Institute Kala Shah Kaku(RRI, KSK), the Rice Research Institute (RRI, Dokri), the Pakistan Agricultural Research Council (PARC, KSK), Guard Rice; Golarchi, Four Brothers Multan, Emaky Sheikhupura, Chaudhry Khair Din (CKD), and Dera GhaziKhan, in the country regarding yield and other quality parameters. Based on in-house trials, three hybrids (HP1, HP2 and HP3) were selected for further evaluation in NUYT and DUS trials. Our hybrids showed excellent performance concerning yield and various quality parameters over check varieties (D-121, Guard-53). The quality traits of seed length (7.01 mm; 6.8 mm), width (2.04 mm; 2.11 mm), thickness (1.82 mm; 1.76 mm), length–width ratio (3.44 mm; 3.22 mm), brown rice (81 gm; 84.4 gm) and milled rice (77.3 gm; 79.4 gm) of HP1 and HP3 were higher than the check varieties. The results showed that the yield kg per hectare of HP1 (8709), HP2 (8833) and HP3 (9338) was more than the check varieties, i.e., D-121 (9171) and Guard-53 (8395). The almost average performance of our hybrid varieties was higher than the check varieties, and all the information was mentioned in Table 3 and Supplementary Table S1.

In 2021, 135 hybrids and two check hybrids were evaluated in ten different locations in Pakistan under adaptability trials. Our hybrids (HP1, HP2 and HP3) showed significant differences to check varieties (D-121 and Guard-53) in yield and various quality parameters (Table 3, Supplementary Table S2). Almost all the traits showed significant variation among each other and with check varieties. The average performance of HP hybrid characteristics was higher than the check varieties in almost all the traits studied. HP1 seed length (7.3 mm), width (2.13 mm), thickness (1.81 mm), length–width ratio (3.43 mm), cooked grain length (12.9 mm), brown rice (82.03 gm), milled rice (76.6 gm) and grain yield 7863 kg/ha was higher than both check hybrids.

The principal components also showed variability among the entire set of genotypes and their contributing traits. Those principal components have more than one eigenvalue that shows more variability and has more importance in selection criteria. In 2020, four components with more than one eigenvalue contributed to variation in a collective 67.43%. Similarly, in 2021, six principal components showed a maximum variation of 82.16%. Such results showed significant differences among genotypes and traits. Some components had significant positive effects on various quality and seed parameters. The components

had more considerable positive effects for the selection of promising genotypes. The eigenvectors with positive values with their respective traits showed more variation in genotypes and studied traits (Table 4).

**Table 4.** Eigenvalue, variation and cumulative% variability of various yield and quality parameters of hybrid rice.

| Traits | PC | 2020 | | | 2021 | | |
|---|---|---|---|---|---|---|---|
| | | Eigenvalue | Variation% | Cumulative % | Eigenvalue | Variation % | Cumulative % |
| Length | PC1 | 2.48 | 22.63 | 22.62 | 3.18 | 22.78 | 22.78 |
| Width | PC2 | 2.13 | 19.45 | 42.07 | 2.60 | 18.58 | 41.36 |
| Thickness | PC3 | 1.49 | 13.56 | 55.63 | 1.92 | 13.72 | 55.08 |
| L/W ratio | PC4 | 1.29 | 11.79 | 67.43 | 1.57 | 11.23 | 66.32 |
| Curling % | PC5 | 0.96 | 8.77 | 76.21 | 1.18 | 8.45 | 74.77 |
| Bursting % | PC6 | 0.71 | 6.46 | 82.67 | 1.03 | 7.38 | 82.16 |
| C.G.L (mm) | PC7 | 0.57 | 5.24 | 87.92 | 0.90 | 6.44 | 88.60 |
| Brown rice % | PC8 | 0.47 | 4.29 | 92.21 | 0.66 | 4.77 | 93.38 |
| Milled rice % | PC9 | 0.39 | 3.63 | 95.85 | 0.51 | 3.69 | 97.08 |
| Head rice % | PC10 | 0.26 | 2.42 | 98.27 | 0.40 | 2.88 | 99.97 |
| Yield/ha | PC11 | 0.18 | 1.72 | 100 | 0.003 | 0.026 | 100 |

Scree plot, biplot and traits variation among the significant principal components numbers (PC1, PC2) had maximum variation between them and in comparison with other numbers. The scree plot describes the association between eigenvalues and cumulative% variability. This showed the variation among genotypes and their traits.

Some of the traits showed positive significant association with each other that was very important for enhancing the various qualitative and quantitative traits of hybrids under the adaptability trials 2020–2021. Seed length had positive significant associations with length–width ratio (r = 0.5159*), thickness (r = 0.3907*) and cooked grain length (r = 0.5027*). A significant positive association was observed between thickness with cooked grain length and yield/hectare (r = 0.4059*; r = 0.3091*). Brown rice was associated with milled rice and head rice in adaptability trials 2020 (Table 5). In 2021, a significant positive association of seed width was observed with length–width ratio, thickness and bursting% (r = 3054*; r =3051*; r = 0.1679*). The bursting%, cooked grain length, brown rice% and yield/hectare had a significant positive association with curling% (Table 6). Correlations among the significant traits were beneficial for selecting better hybrids and determining the interrelationship between the traits. According to the National Uniform Yield Trial (NUYT), Distinctness, Uniformity and Stability (DUS) confirmed that HP1 and HP3 hybrid varieties performed very well in both years over check varieties (D-121, Guard-53) in various quality and yield parameters. In the first year of the trials, seven different sites were selected for testing the hybrid varieties, and in the second year, ten different sites were selected for testing the hybrids. In all locations, HP hybrids showed excellent results in all country zones.

**Table 5.** Association of various morphological, quality and yield traits of hybrid rice in adaptability trials 2020.

| Variables | Length (mm) | Width (mm) | L/W Ratio | Thickness (mm) | Curling (%) | Bursting (%) | C.G. L (mm) | Brown Rice (%) | Milled Rice (%) | Head Rice (%) | Yield/ha |
|---|---|---|---|---|---|---|---|---|---|---|---|
| Length (mm) | 1.00 | | | | | | | | | | |
| Width (mm) | 0.0961 | 1.00 | | | | | | | | | |
| L/W Ratio | 0.5159 * | 0.2458 * | 1.00 | | | | | | | | |
| Thickness (mm) | 0.3907 * | −0.2190 * | −0.2532 * | 1.00 | | | | | | | |
| Curling (%) | −0.1223 | −0.0416 | 0.1757 | −0.3379 * | 1.00 | | | | | | |
| Bursting (%) | −0.1830 | −0.1154 | −0.1700 | −0.0676 | 0.3539 * | 1.00 | | | | | |
| C.G. L (mm) | 0.5027 * | 0.0173 | 0.2155 * | 0.4059 * | −0.2315 * | −0.1600 | 1.00 | | | | |
| Brown Rice (%) | −0.0942 | −0.1156 | −0.0335 | 0.0936 | 0.0885 | 0.1505 | −0.1284 | 1.00 | | | |
| Milled Rice (%) | −0.0687 | −0.2450 * | −0.1475 | 0.1706 | 0.0446 | −0.0940 | −0.0961 | 0.6637 * | 1.00 | | |
| Head Rice (%) | −0.2902 * | −0.1262 | −0.1737 | −0.0459 | −0.0626 | −0.1414 | −0.1488 | 0.4243 * | 0.4807 * | 1.00 | |
| Yield/ha | 0.0122 | −0.0263 | −0.2354 * | 0.3091 * | −0.0846 | −0.0175 | 0.1135 | 0.1474 | 0.1423 | −0.0296 | 1.00 |

* showed the Level of significance. $p < 0.05$= * (Statistically significant) and $p$ represent the probability value.

**Table 6.** Association of various morphological, quality and yield traits of hybrid rice in adaptability trials 2021.

| Variables | Length (mm) | Width (mm) | L/W Ratio | Thickness (mm) | Curling (%) | Bursting (%) | C.G. L (mm) | Brown Rice (%) | Milled Rice (%) | Head Rice (%) | Yield/ha |
|---|---|---|---|---|---|---|---|---|---|---|---|
| Length (mm) | 1.00 | | | | | | | | | | |
| Width (mm) | −0.6068 * | 1.00 | | | | | | | | | |
| L/W Ratio | −0.3201 * | 0.3054 * | 1.00 | | | | | | | | |
| Thickness (mm) | −0.3201 * | 0.3051 * | 1.0000 * | 1.00 | | | | | | | |
| Curling (%) | 0.0586 | 0.1062 | −0.1481 | −0.1482 | 1.00 | | | | | | |
| Bursting (%) | −0.0227 | 0.1679 * | −0.1625 | −0.1626 | 0.4136 * | 1.00 | | | | | |
| C.G. L (mm) | 0.0576 | −0.1179 | 0.0670 | 0.0670 | 0.1951 * | −0.8077 * | 1.00 | | | | |
| Brown Rice (%) | 0.1666 | 0.0248 | −0.1328 | −0.1328 | 0.5975 * | 0.1447 | 0.2168 * | 1.00 | | | |
| Milled Rice (%) | −0.1234 | 0.1154 | 0.1646 | 0.1646 | −0.1375 | −0.2631 * | 0.1943 * | −0.1055 | 1.00 | | |
| Head Rice (%) | 0.0471 | −0.0213 | −0.0600 | −0.0600 | −0.0176 | −0.1663 | 0.1665 | −0.0196 | 0.4282 * | 1.00 | |
| Yield/ha | −0.0831 | 0.0222 | 0.0193 | 0.0193 | 0.1981* | 0.2262* | −0.1145 | 0.0637 | −0.1348 | −0.0656 | 1.00 |

* showed the Level of significance. $p < 0.05$= * (Statistically significant) and $p$ represent the probability value.

### 3.4. DNA Analysis

Following the SSR marker method for identifying rice varieties (NY/T1433-2014), an experiment was conducted to analyze HP1, HP2 and HP3 hybrid rice varieties; a total of 48 primer pairs were used for this purpose. Among them, nine pairs of markers could clearly distinguish the three varieties and showed more remarkable genetic diversity (Figure 2). No.1, No.2, No.3, No.4, No.5, No.6, and No.7 primers distinguished HP1 from the other two varieties (HP2, HP3). On the other hand, No.2 and No.5 primers distinguished HP2 from the other two varieties (HP1, HP3) and No.5, No.8, and No.9 primers distinguished HP3 from the other two varieties (HP1 and HP2).

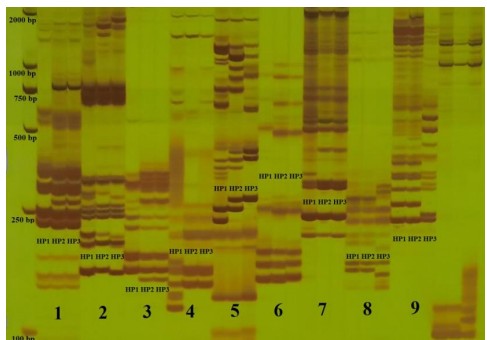

**Figure 2.** Genetic diversity of elite Honglian type of hybrid rice varieties.Markers (RM219, RM236, RM274, RM253, RM424) distinguished HP1 with HP2 and Hp3. Markers RM236 and RM424 distinguished HP2 with HP1 and HP3. Markers RM424, RM481 and RM493 distinguished HP3 with HP1 and HP2.

## 4. Discussion

### 4.1. Genetic Studies and Characteristics of HonglianType Hybrid Rice

Crop improvement is followed by genetic diversity, with various characteristics screened. In this study, different hybrids were studied in 2019 under in-house trials for yield and yield-related traits and their adaptation to changing climatic conditions. With respect to yield, all the Honglian hybrid rice varieties showed more than 30% more yield potential over check variety. As Honglian hybrid rice varieties have desirable yield and yield related traits, i.e., high yield, high quality, heat tolerant, drought tolerant, high tillering ability, high fertility and good grain quality parameters, etc., PCA and correlation analysis displayed significant genetic differences among the genotypes along with all the desired traits indicating the existence of variability [28,29]. The association of different rice traits and patterns of influence on the grain yield of rice was investigated. Such types of evaluation are very important to determine the direct effects of various traits on yield to determine the selection criteria for high grain yield. We found that some of traits have greater value, including seed length, seed width, seed thickness, curling%, bursting %, cooked grain length, head rice recovery %, etc., which have accounted for high grain yield.The positive associations of yield with other desirable traits were found to be significant, providing the information for selecting desirable rice hybrids/genotypes which are more favorable to acclimatize to changing environments. Positive significant genetic differences and correlation studies among the genotypes, along with desired traits, provide information to the researcher for the better selection of genotypes [30,31].

Genetic variation of traits is essential for selection and other breeding applications [32]. Similarly, in previous studies, the Honglian type of hybrid rice displayed good performance in various Southeast Asian countries on the basis of various desirable traits [33].The rice hybrids showed significant differences on the basis of various morphological traits.It was found that the average performance of HP1 and HP3 was excellent over check varieties in both years with respect to yield and yield related traits.An NUYT test provides excellent information regarding the adaptability of HP hybrids over a wide range of environments in all the locations of Pakistan, and is also helpful for the food security of China and the world's population due to its high yield and other desired traits [34]. This was a significant step towards selecting good hybrids in different ecological zones, providing information regarding the suitability of hybrid seed production and technology transfer of Honglian hybrid rice to farmers, students and scientists. With the advancement of new molecular breeding techniques and genomics technology, two-line hybrid CMS and three-line hybrid CMS systems can be adopted to improve crops. Hence, the CMS systems of hybrid rice have great scope and are helpful for the enhancement of productivity [35,36]. Molecular markers are significant for determining genetic diversity and developing new hybrids [37,38]. DNA fingerprint results showed that Honglian type varieties are highly variable from the traditional indica rice, which will support as an alternative germplasm for breeding selection, which is useful for modifying the local varieties in Pakistan. Some of the markers have been used in previous studies to determine the genetic variation among the Honglian type of hybrid rice and their genetic traits [37].

### 4.2. Correlation Studies

Correlation analysis allows us toobtain information on the relationship between variables, i.e., dependent and independent. It helps plant breeders to better understand the relationship of the yield-related traits, which will lead to the selection of genotypes with desired characteristics [39]. Knowledge about correlation coefficients is essential because the grain yield and other quantitative traits are influenced by many factors [40,41]. Correlation is one of the best methods to find out the relations among various traits and to lead the way for the frequency of traits and the compulsory screening to be measured in improving traits, for instance, grain yield [31,42].

### 4.3. Principal Component Analysis (PCA) Studies

The hybrid varieties were also analyzed using PCA to compare characteristics of HP varieties over check varieties. Principal components with more than one eigenvalue showed more variability among the traits studied for each genotype. Those principal components with more than one eigenvalue showed a collective variation of 67.42% in the year 2020. The PC1 had 22.62%, PC2 showed 19.45%, PC3 exhibited 13.56%, and PC4 had 11.79% variability between the rice varieties and their various traits in 2020. The variance and eigenvalue associated with principal components decreased gradually and stopped at 0.18%. In the year 2021, the first six components showed maximum variability of 82.13% (Table 4).

Based on the principal eigenvalue components, variation is considered to be very important in screening and selecting the rice hybrids/varieties [43]. Principal components and eigenvalues that showed genetic differences among genotypes are shown in Figure 3. Higher eigen values showed more variability and helped to select parents and improve the varieties through breeding [44]. In biplot principal component analysis, almost all traits positively affected their respective genotypes except curling% and bursting % (Figure 4). This showed the diversity of the genotypes, along with the desired characteristics. This information will be beneficial for the further screening of genotypes for developing a new plant population and starting a new breeding program. In adaptability trials, traits showed variability with the positive effect and their principal components (Figure 5), which were considered more important in selecting diverse genotypes. Such genotypic and phenotypic characteristics could be utilized in a breeding program for the screening and developing of new plant populations [45,46]. The study was equally beneficial for the best interests of breeders and researchers to utilize in their research in the best interest of the community and economy of the country.

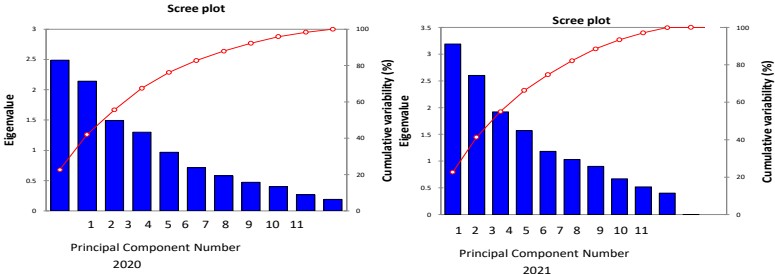

**Figure 3.** Scree plot of principal component analysis rice hybrids between their eigen values, number of principal component and accumulative variability% under adaptability trials 2020–2021.

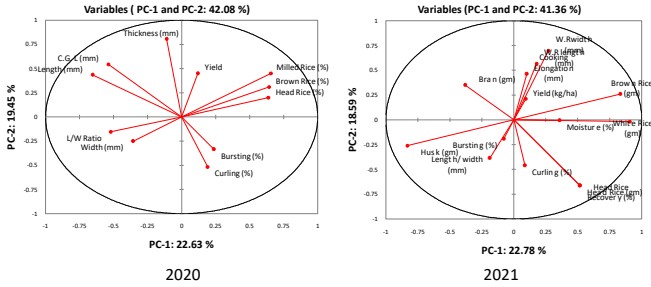

**Figure 4.** Traits variation on the basis of their major principal components under adaptability trials 2020–2021.

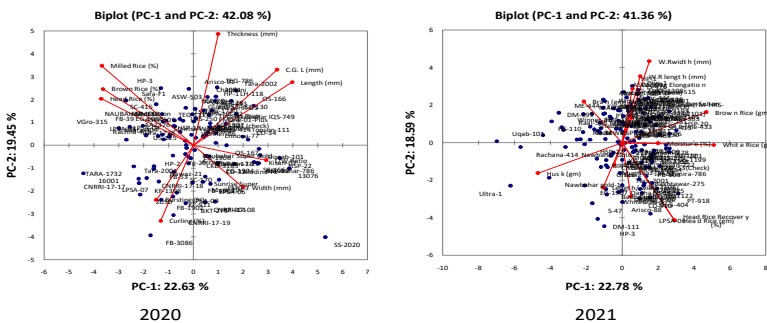

**Figure 5.** Biplot PCA analysis of various hybrid rice varieties and their traits association with principal component numbers under adaptability trials 2020–2021.

Honglian hybrid rice (HP1, HP3) yield is higher than the check variety in continuous year inspections and different locations, and its yield is higher because it is heat tolerant and widely adaptive, so the yield is comparably stable in different environments. Moreover, its seeds setting rate, tiller numbers, and spikes per plants are higher than the other varieties, which is the solid base for the higher yield, and this is critical for the agriculturally safe production in Pakistan, especially facing the challenges of climate change and sharp environmental variations in the future.Exploiting the heterosis of the hybrids had great importance for improving high grain yield in rice and other traits [47]. The parental material of Honglian hybrid rice had a strong restoration ability in the cytoplasmic male sterility system (CMS) for developing new three-line hybrids [48–51].

### 4.4. Honglian Type Hybrid Rice Research Importance and Future Prospects

"Food and safety come as the first". Rice is the staple food for more than 50% of the world population and its demand is increasing day by day. Therefore, to gradually improve wide-ranging rice production capacity and meet the inflexible demands for food consumption globally is a major strategic issue concerning food security, social stability, people's health, and economic developments. Wuhan University is one of the leading teams in the field of hybrid rice research in China. The team made many research achievements and developed a series of Honglian type hybrid rice varieties with excellent characteristics such as high quality, high yield, wide adaptability, multi-diseases/insect resistance and environmental friendliness. The main characteristics are prominent genetic diversity, wide hybridization selectivity and excellent comprehensive agronomic traits [52].

Several novel techniques could be beneficial for improving hybrid rice breeding techniques for developing new hybrid varieties. To advance the application of hybrid rice, genome editing technology based on clustered regularly interspaced palindromic repeats (CRISPR), the Cas9 system, has been extensively used to improve crops [53,54]. Another technique, de novo domestication of allotetraploid rice, could be very fruitful to stabilize heterosis and other desirable traits in hybrid rice which show more adaptation in changing environments, such asin other polyploidy species, i.e.,wheat, triticale, cotton, tobacco and strawberry [55–57]. Genetic resources in hybrid rice breeding will be very desirable in the future to develop breeding techniques for improving crops and enhancing the selection process of superior hybrid lines.

In this regard, the University of the Punjab and Wuhan University, China have plans for the further testing of Honglian-type hybrid rice varieties and hybrid seed production in Pakistan under different climatic conditionsto achieve maximum results in the rice field through technology transformation techniques. Different plant populations ($F_2$, RILs, NILs, etc.) will be developed by using diverse germplasm (Figure 6) to produce elite breeding plant varieties to meet the food requirements of the world's population.

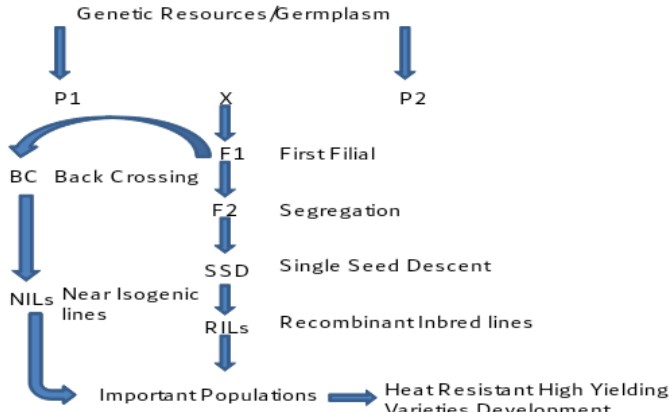

**Figure 6.** Genetic resources are a vital component for the improvement and development of new hybrid rice high-yielding varieties.

## 5. Conclusions

The present findings conclusively demonstrate that the yield of 9338 kg/ha of HP3 in 2020 and 7863 kg/ha of HP1 in 2021 were higher than the average yield of all the hybrids tested in both years and over check varieties. We found that some of the traits, i.e., seed length, width, thickness, cooked grain length, brown rice, milled rice and yield per hectare of HP3 and HP1, respectively, were greater than the check varieties (D-121, Guard-53). These hybrids and their parent material could be further used for the development of new distinct uniform homozygous plant populationson the basis of the desired characteristics. The performance of Honglian type hybrid rice was more stable and considerably better than other hybrids/varieties in high temperature locations, which indicates its better adaptation and acclimatization in various ecological zones of Pakistan. Finally, we summarize that HP1,HP2 and HP3, as the new generations of Honglian type hybrids with advanced characteristics, are beneficial for introduction into Pakistan for future development and industrialization. The NUYT and DUS analysis convincingly showed that they are suitable to grow in Pakistan. Based on the historical economical contributions of the old varieties of Honglian type hybrid rice, it is promising that development and industrialization of these new varieties would contribute well to Pakistan and benefit the people both in China and Pakistan. In the current scenario, this type of study and the genetic material could be very useful for the production of high yielding varieties that would be more fruitful to the farmers community and strengthen the country's economy.

**Supplementary Materials:** The following are available online at https://www.mdpi.com/article/10.3390/agriculture13020242/s1, Table S1: (Part-I): Means of Paddy Yield (kg/ha) of Rice Hybrids Evaluated in NUYT during Kharif, 2020, 2020, (Part-II). Quality data of NUYT 2020 Hybrids Rice; Table S2: (Part-I): Mean of Paddy Yield (kg/ha) of Rice Hybrids Evaluated in NUYT during Kharif, 2021, (Part-II) Quality Characteristics of Rice Hybrids Evaluated in NUYT 2021; Table S3: SSR markers used for rice hybrids for their Genetic diversity study.

**Author Contributions:** M.A. (Muhammad Ashfaq) conceived the idea of the study and wrote up the manuscript. R.Z. is the principal researcher and breeder of the Honglian type hybrid rice and provided the material. Z.X. carried out the SSR marker analysis. M.A. (Muhammad Ali) and A.R. helped to organize the experimental material for further data recording. M.J. and A.S. helped with data analysis and supported the write-up and review of the manuscript. X.W. designed the experiment and helped a lot in the completion/write-up of the research work. All authors have read and agreed to the published version of the manuscript.

**Funding:** This research was funded by the Ministry of Science and Technology (MOST), China (National Key Research and Development Project: 2021YFE0101000) and the Pakistan Science Foundation (PSF/CRP/18th Protocol (11) under the international collaboration project supported by both governments. The APC was funded by the collaborative project.

**Institutional Review Board Statement:** Not applicable.

**Informed Consent Statement:** Not applicable.

**Data Availability Statement:** The data that support the findings of this are available in the main manuscript and the supplementary files.

**Acknowledgments:** The authors would like to thank the Wuhan University, China and the University of Punjab for supporting and facilitating the joint project on Honglian-type hybrid rice. The Ministry of Science and Technology of both countries and the Pakistan Science Foundation (PSF) supported financially to complete this research work in a timely manner.

**Conflicts of Interest:** The authors declare no conflict of interest.

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
