# Peer review of "Adaptation and High Yield Performance of Honglian Type Hybrid Rice in Pakistan with Desirable Agricultural Traits"

_agriculture, doi:10.3390/agriculture13020242_

Round 1
Reviewer 1 Report (Previous Reviewer 1)
Dear Authors and Editor
The article only corrects minor errors compared to the previous version. The main issues that the reviewer commented on last time were barely edited.
Here is my previous comments for the first version, and compare to the second version, no improvements were found.
=====================
01- This manuscript is poorly written, not meeting the minimum standards of a scientific paper. There are more personal feelings in presenting and analysing the results. This should be avoided.
This seems to be a test report of 3 hybrid rice strains HP1, HP2 and HP3 according to the "National Uniform Yield Trials" and "Distinctness, Uniformity and Stability" according to the requirements of Pakistan Government; the author then upgraded and interpreted these three hybrid rice strains as ecologically suitable in Pakistan and even claimed to be "Moreover, it was considerably better, suggesting that Honglian-type hybrid rice varieties of Pakistan is less risky under climate change and global warming conditions"
This manuscript seemed numerous points that need correction and revision for clear understanding.
02- The abstract is too long. Please rewrite. Some formats and detail of the manuscript may not be MDPI format. The title is not related to the detail of the results. Please clearly specify the highlight of this study.
03- The introduction is nonsense, not a scientific paper. No introduction of hybrid rice varieties in this study, their genetic and agronomic characteristics, or their origin, ...
What is the objective of this study? The introduction should be included 1) What is known 2) what is unknown / problem / Research GAP and 3) Hypothesis / how to solve this problem or close GAP.
Please more introduction about the Gap between the previous study and the importance of this study [the content in the discussion should be filtered and moved to the introduction part].
Please add more information in the introduction part. And indicate the objective of this study at the end of the introduction part.
04- The Method and material part: Too bad, too superficial and not enough information to repeat the experiments or verify.
- No one knows the profiles of hybrid rice strains (HP1, HP2, HP3, HP4, HP5, HP6, HLR-006, 105 WR-1906, Guard53, D-121 and KSK-13) used for this study. Please provide their full profiles such as their genetic, agronomic characteristics, and origin?
- Experimental location? In-house or in-field? Scale (m2 or ha?). Time of study, detailed planting start time and harvest time.
- Lines 137 - 139 page 4 "A total of 104 hybrids in 2020 and 137 hybrids in 2021 (including Honglian hybrid rice and check varieties) were tested under different ecological zones of Pakistan", please detail the 104 and 137 hybrid strains that were used in this study? Meanwhile, please specify the ecological parameters of these eco-zones; no one knows what are the ecological niches in this study. Otherwise, no one knows why (or how) the difference in ecological zones in Pakistan affected the yield performance of the experimental hybrid rice strains.
05- The Results and Discussion: Too bad; can't read. The structure, writing style, comparative presentation and argumentation of the author are too poor.
- Table 1 does not have information on the location, size and duration of the trial
- Table 2 is not the result of this study, these primers are collected from one or more different sources. The author does not understand that it should be in the Method Materials (or appendix) but why is the result part?
- Page 6, lines 196-197, "RRI, KSK; RRI, Dokri; PARC, KSK, Guard; Golarchi, Four Brothers Multan, Emaky Sheikhupura, CKD, DG Khan" what are they? The readers can not understand.
- The same for page 7, line 211
Almost results show only results (that cannot see in the figure or table), please explain the meaning of the results for clearly understood. Please try to design the results to meet the objective or related to the explanation in the text.
When discussed and compared with other research, how can compare directly? Please compare the meaning of the results, do not compare the results.
In the discussion part, please truly discuss, do not review or introduce again. (I recommend the author filter the details in the discussion part and then move to the introduction part).
06-The conclusions are so strange due to the objective of this study is not clear. The author's conclusion rambled, "It was also observed that under higher temperatures, the performance of Honglian type hybrid rice was stable and considerably better, which suggests that the cultivation of Honglian type hybrid rice varieties in Pakistan is less risky under the scenario of climate change and global warming due to high fertility rate. Such genetic material could be very useful as genetic resources for developing new elite breeding lines in changing ecosystems. The study will be equally beneficial to the farmers and the scientific community to strengthen the country's economy"
But none of the results presented in the manuscript support this claim.
Last but not least: This article is intended to promote the Honglian hybrid rice line, rather than as a research study that could provide the scientific community with meaningful results.
=====================
Best wishes
Author Response
Dear Editors and Reviewers of Agriculture,
Thank you very much for considering our manuscript entitled "Adaptation and High yield performance of Honglian type hybrid rice in Pakistan with desirable agricultural traits" (Manuscript ID: agriculture-2037805). We have carefully revised the manuscript according to the reviewers’ comments and editorial evaluation. Our point-by-point responses to the reviewers’ comments are presented below.
Dear Authors and Editor
The article only corrects minor errors compared to the previous version. The main issues that the reviewer commented on last time were barely edited.
Here is my previous comments for the first version, and compare to the second version, no improvements were found.
Response: Dear reviewer thank you very much for your valuable comments for the improvements of the article. I tried my best to improve article as per your suggestions. All the improvements/corrections are highlighted in green colour in the main manuscript. Please see the point by point response here.
Reviewer #1
- This manuscript is poorly written, not meeting the minimum standards of a scientific paper. There are more personal feelings in presenting and analysing the results. This should be avoided.
This seems to be a test report of 3 hybrid rice strains HP1, HP2 and HP3 to the "National Uniform Yield Trials" and "Distinctness, Uniformity and Stability" according to the requirements of Pakistan Government; the author then upgraded and interpreted these three hybrid rice strains as ecologically suitable in Pakistan and even claimed to be "Moreover, it was considerably better, suggesting that Honglian-type hybrid rice varieties of Pakistan is less risky under climate change and global warming conditions"
This manuscript seemed numerous points that need correction and revision for clear understanding.
Response: Thanks for very constructive comment. Firstly, different hybrids were tested under field conditions in various locations of Punjab, Pakistan on the basis of their morphological characters in the year 2019-20. From them three hybrids (HP1, HP2, HP3) were selected for further evaluation in DUS and NUYT trials on the basis of desirable traits in all over Pakistan of the designated locations of the Government of Pakistan. At some locations the temperature was higher more than 40°C e.g. Golarchi Larkana Sindh at the time of fertility the temperature was around about 45°C. Honglian type of hybrid rice varieties performed very well in such locations due to its excellent characteristics and heat tolerance ability, that could be very useful in climatic changing environment. “Moreover, it was considerably better, suggesting that Honglian-type hybrid rice varieties can be grown in Pakistan because they are less risky under climate changes, especially by the global warming challenges.” Page No2 and Line No45-47.
- The abstract is too long. Please rewrite. Some formats and detail of the manuscript may not be MDPI format. The title is not related to the detail of the results. Please clearly specify the highlight of this study.
Response: Thanks for the valuable comments. As per instruction the abstract has been modified and improved. Result section has been improved and highlighted the specific objectives of the study according to your suggestion (Line No16-18).
- The introduction. No introduction of hybrid rice varieties in this study, their genetic and agronomic characteristics, or their origin.
What is the objective of this study? The introduction should be included 1) What is known 2) what is unknown / problem / Research GAP and 3) Hypothesis / how to solve this problem or close GAP.
Please more introduction about the Gap between the previous study and the importance of this study [the content in the discussion should be filtered and moved to the introduction part.
Please add more information in the introduction part. And indicate the objective of this study at the end of the introduction part.
Response: Thanks for the very constructive comment. As per instruction/suggestion the introduction part has been improved. The information regarding previous study and current study has been added and all the corrections are highlighted in green colour. At the end of introduction part the specific objectives of this study has been included (Line No 81-97 and line No107-113).
- The Method and material part: Too bad, too superficial and not enough information to repeat the experiments or verify.
No one knows the profiles of hybrid rice strains (HP1, HP2, HP3, HP4, HP5, HP6, HLR-006, 105 WR-1906, Guard53, D-121 and KSK-13) used for this study. Please provide their full profiles such as their genetic, agronomic characteristics, and origin?
Experimental location? In-house or in-field? Scale (m2 or ha?). Time of study, detailed planting start time and harvest time.
Lines 137 - 139 page 4 "A total of 104 hybrids in 2020 and 137 hybrids in 2021 (including Honglian hybrid rice and check varieties) were tested under different ecological zones of Pakistan", please detail the 104 and 137 hybrid strains that were used in this study? Meanwhile, please specify the ecological parameters of these eco-zones; no one knows what are the ecological niches in this study. Otherwise, no one knows why (or how) the difference in ecological zones in Pakistan affected the yield performance of the experimental hybrid rice strains.
Response: Thanks for very constructive comment. The materials and methods section has been improved as per instruction. The profiles of the hybrid have been added in the manuscript with all other details regarding experimental location, sowing and harvesting date and other characteristics of the hybrids. The detailed information regarding hybrid characteristic and other informations have been given in the supplementary files (Line No147-161), table2. The detailed information has been given in supplementary table1 and table 2.
- The Results and Discussion: can't read. The structure, writing style, comparative presentation and argumentation of the author are insufficient.
Table 1 does not have information on the location, size and duration of the trial.
Table 2 is not the result of this study, these primers are collected from one or more different sources. The author does not understand that it should be in the Method Materials (or appendix) but why is the result part?
Page 6, lines 196-197, "RRI, KSK; RRI, Dokri; PARC, KSK, Guard; Golarchi, Four Brothers Multan, Emaky Sheikhupura, CKD, DG Khan" what are they? The readers can not understand.
The same for page 7, line 211.
Almost results show only results (that cannot see in the figure or table), please explain the meaning of the results for clearly understood. Please try to design the results to meet the objective or related to the explanation in the text.
When discussed and compared with other research, how can compare directly? Please compare the meaning of the results, do not compare the results.
In the discussion part, please truly discuss, do not review or introduce again. (I recommend the author filter the details in the discussion part and then move to the introduction part.
Response: Thanks for very valuable comments. The results and discussion section has been improved as per instruction. The information in the table1 has been improved and table2 shifted into materials and methods as per suggestion line No 234-235. The full names of different locations have been added in the main manuscript line No202-211 and 288-292. The results section has been improved and modified as per instruction. Comparative information has been added in discussion section with respect to previous study. Discussion and introduction part has been filtered according to instruction/suggestion. Line No73-87.
06-The conclusions are so strange due to the objective of this study is not clear. The author's conclusion rambled, "It was also observed that under higher temperatures, the performance of Honglian type hybrid rice was stable and considerably better, which suggests that the cultivation of Honglian type hybrid rice varieties in Pakistan is less risky under the scenario of climate change and global warming due to high fertility rate. Such genetic material could be very useful as genetic resources for developing new elite breeding lines in changing ecosystems. The study will be equally beneficial to the farmers and the scientific community to strengthen the country's economy" Response: Thanks for very constructive comment. As per instruction the information has been improved and incorporated in the conclusion section according to the suggestions Line No 629-650.

Reviewer 2 Report (Previous Reviewer 2)
The authors brought changes in their manuscript, but not as much as it required. All of my comments from the previous reviewer process are not considered. The authors should consider my comments or they should respond them.
Author Response
Dear Editors and Reviewers of Agriculture,
Thank you very much for considering our manuscript entitled "Adaptation and High yield performance of Honglian type hybrid rice in Pakistan with desirable agricultural traits" (Manuscript ID: agriculture-2037805). We have carefully revised the manuscript according to the reviewers’ comments and editorial evaluation. Our point-by-point responses to the reviewers’ comments are presented below.
Reviewer #2
The authors brought changes in their manuscript, but not as much as it required. All of my comments from the previous reviewer process are not considered. The authors should consider my comments or they should respond them.
The manuscript entitled “Adaptation and High yield performance of Honglian type hybrid rice in Pakistan with excellent agricultural traits” submitted to agriculture needs corrections and improvements. Generally, the manuscript has lack of scientific contribution and written complicated. Anyway, below are major and minor comments for the improvement of the manuscript.
Response: Thanks for your critical review and good comments.
- English language revision is needed.
Response: Thanks for pointing the mistakes. I have carefully proofread the manuscript and correct the linguistic mistakes and all the corrections have been made in blue color.
- It will be better to select key words that are not mention in the title.
Response: Thanks for the comments. I have mentioned as per instruction Line No48-49.
- L13-14: remove the repeated work (high).
Response: Thanks for pointing the mistake. I have removed the repeated work and correct as per instruction.
- L27: explain PCA. The first abbreviation should be explained. Line No30 in Abstract
Response: As per instruction abbreviation has been rewrite in full form.
- The manuscript has lack of a clear objective.
Response: The objective has been incorporated in the manuscript as per suggestion/instruction. Line No16-18 in abstract.
- What is the difference between the traits in L139 and L143 (length, wide, thickness and so on).
Response: Thanks for pointing the mistakes. Corrected as per instruction Yield is quantitative trait and all other traits are quality traits. Line 163-167.
- The varieties selection is somehow complicated. The authors selected ten hybrid rice varieties but presented only 4 varieties in Table 1. However, 3 hybrids with 2 check varieties are compared in Table 3.
Response: On the basis of seed morphological parameters performance the three hybrids were selected from ten hybrids. Firstly, the selected hybrids were compared with one check variety in-house trials. Secondly, these hybrids were tested in all over Pakistan in different locations along with check varieties and more than 100 other hybrid rice varieties. Other varieties data has been mentioned in the table 2 line no 255-256.. Regarding table 3 only presented the summarize data of Honglian type of hybrid rice. Please see the detail information in supplementary table1 and table2..
- The authors showed PCA analysis in Table 4. It is unnecessary to present the data in correlation coefficient tables too, as presented in Tables 7-8.
Response: Thanks for the comment. As per instruction the unnecessary tables 7-8 has been removed from the manuscript (Line No371-372).
- The authors presented the results of PCA in Table 4 but repeated in Figures 3, 4 and 5.
Response: Thanks for the comment. The data was presented separately for more clarity of the traits, variable observations and response of each trait with respect to their positive and negative effects.
- What does figure 6 shown?
Response: Thanks for the comment. This is also part of this manuscript just to show the procedure for the development of honglian hybrid rice elite plant population and also with respect to future prospects for the development of mapping populations (RILs, NILs, F2).
- Citation must follow the format of journal.
Response: Thanks, as per suggestion for citation journal instructions, all citations has been uniformed followed the journal formatting requirements.
- The conclusion section should be rewritten following the important points of the manuscript.
Response: Thanks for the comment. The conclusion section has been improved as per instruction (Line No.595-605.
- Check the references thoroughly and unify them based on the guidance of the journal.
Response: Thanks for the comment. All the references were thoroughly checked based on the guidance of the journal.

Reviewer 3 Report (Previous Reviewer 3)
The revised version of the manuscript has been significantly made. However, the current form of the paper still needs some major revisions. The authors can find some useful comments and suggestions (see in pdf file). it may be useful to improve the quality of the manuscript.
1. The abstract should be reworded (see comments in pdf file
2. Materials and Methods must be added some important information, for instance, Must be mentioned the check varieties and why used them.
3. The subsection " Traits measurement" the unit used for the equation must be mentioned.
4.This Fig is not clear, if possible, change by another one with a high solution
5. Where were the in-house trial methods? And where was the data,
6. The DNA fingerprinting and PCR analysis should be added some important information such as PCR and electrophoresis.
7. Statistical analysis should be added information on the significant value
8. In the result part: Which places were done in the University of Punijab and Wuhan University China, specifically classify!
9. Section 3.4 DNA analysis should be reworded, his Fig is unclear which marker (name of marker) is polymorphic? Should take note below Fig, what are the lanes 1-7.
10. Recheck all spelling, typing and grammar errors (see in the file), all references must be rechecked and formatted following the submitted journal style.

Author Response
Dear Editors and Reviewers of Agriculture,
Thank you very much for considering our manuscript entitled "Adaptation and High yield performance of Honglian type hybrid rice in Pakistan with desirable agricultural traits" (Manuscript ID: agriculture-2037805). We have carefully revised the manuscript according to the reviewers’ comments and editorial evaluation. Our point-by-point responses to the reviewers’ comments are presented below.
Reviewer #3
The revised version of the manuscript has been significantly made. However, the current form of the paper still needs some major revisions. The authors can find some useful comments and suggestions (see in pdf file, it may be useful to improve the quality of the manuscript.
Response: Dear reviewer thank you very much for your very constructive comments for the improvement of the article. As per suggestions the article has been improved with respect to all the sections. Please see the point by point responses.
- The abstract should be reworded (see comments in pdf file
Response: Thank you very much for the constructive comments. As per suggestion and according to the pdf file, the corrections have been made in the main file and all the corrections highlighted in red color.
- Materials and Methods must be added some important information, for instance, Must be mentioned the check varieties and why used them.
Response: The information has been added in the main file as per suggestion (Line no 145-151)
- The subsection " Traits measurement" the unit used for the equation must be mentioned.
Response: The millimeter unit has been mentioned in the main file (Line No168)
4.This Fig is not clear, if possible, change by another one with a high solution
Response: The figure has been changed as per suggestion (Line 182-183)
- Where were the in-house trial methods? And where was the data,
Response: In house trials the Randomized Complete Block Design (RCBD) design was used and yield parameters were studied. The data was taken in Pakistan. Yield data were mentioned in the table 2 (Line 254-255)
- The DNA fingerprinting and PCR analysis should be added some important information such as PCR and electrophoresis.
Response: The information regarding PCR and electrophoresis has been added in the main file (Line No222-229)
- Statistical analysis should be added information on the significant value
Response: Statistical significance is determination of relationships between two or more variables at level of significance of 1% and 5%. (Line No241-242)
- In the result part: Which places were done in the University of Punijab and Wuhan University China, specifically classify!
Response: The correction has been made in the main file. All the traits were studied in Pakistan at different locations of Pakistan. In-house trials were done in one location which was selected at Faculty of Agricultural Sciences, University of the Punjab, Lahore and other two locations (Gujranwala, Pakpattan) Line No248-249.
- Section 3.4 DNA analysis should be reworded, his Fig is unclear which marker (name of marker) is polymorphic? Should take note below Fig, what are the lanes 1-7.
Response: The information has been added in the main manuscript as per suggestion. A total of 7 pairs of markers could clearly distinguish the three varieties. No.1, No.2, No.3, No.4, No.5 could separate HP1 from the other two varieties, No.2, No.5 could separate HP2 from the other two varieties, No.5, No.8, No.9 could separate HP3 from the other two varieties. A note has been added at the end of the figure.
- Recheck all spelling, typing and grammar errors (see in the file), all references must be rechecked and formatted following the submitted journal style.
Response: Grammatical errors have been removed as per suggestion and rechecked all the references according to Journal style

Round 2
Reviewer 1 Report (Previous Reviewer 1)
Dear Authors
The second version is better the first one and fulfilled the requirements of scientific article journal.
Congratulation.
Author Response
Dear Reviewer and Editor of Agriculture,
Dear reviewers, thank you very much for your valuable and constructive comments for the improvements of the article. We have improved all the section of the manuscript as per your suggestions. Thanks once again for your nice comments for the improvements of the article. All the improvements/corrections are highlighted in green color in the main manuscript.
Reviewer#1
Thank you very much for considering our manuscript entitled "Adaptation and High yield performance of Honglian type hybrid rice in Pakistan with desirable agricultural traits" (Manuscript ID: agriculture-2037805). We have carefully revised the manuscript according to the reviewers’ comments and editorial evaluation.
Dear Authors
The second version is better the first one and fulfilled the requirements of scientific article journal.
Congratulation.
Response: Thank you for your positive evaluation for our manuscript.

Reviewer 2 Report (Previous Reviewer 2)
The manuscript entitled “Adaptation and High yield performance of Honglian type hybrid rice in Pakistan with excellent agricultural traits” submitted to agriculture needs corrections and improvements. Generally, the manuscript is improved but still there are several problems with this manuscript. Below are major and minor comments for the improvement of the manuscript.
1. L39: HP3 yield was higher than over check varieties (D-121, Guard-53). How about HP1 and HP2?
2. L40: HP1 yield (7863 kg/ha) was higher than the average over-check varieties/hybrids. How about HP2 and HP3?
3. The text in the introduction is too long with unnecessary sentences. I recommend that the authors should shorten it with the relevant sentences and paragraphs to this work.
4. The main objective of the study was to screening, genetic diversity, adaptation and high yield performance of honglian type hybrid rice in Pakistan. Please rephrase this sentence.
5. “A total of 104 hybrids in 2020 and 137 hybrids in 2021 (including Honglian hybrid rice and check varieties) were tested under different ecological zones of Pakistan”. This paragraph is repeated in the L178-179.
6. L268: Statistical significance is determination of relationships between two or more variables at level of significance of 1% and 5%. Rewrite this sentence.
7. Where is the data for section 3.2?
8. What does figure 6 shown?
9. I recommend that the authors should carefully check the manuscript. Present their work clearly and concisely. Select the tables and figures which are very important and give values to their text.
Author Response
Reviewer#2
Dear Reviewer and Editor of Agriculture,
The manuscript entitled “Adaptation and High yield performance of Honglian type hybrid rice in Pakistan with excellent agricultural traits” submitted to agriculture needs corrections and improvements. Generally, the manuscript is improved but still there are several problems with this manuscript. Below are major and minor comments for the improvement of the manuscript.
Response: Thanks for your good comments to help us improving the manuscript. Please see point by point corrections as per comments and instructions. All the corrections have been made in blue color.
- L39: HP3 yield was higher than over check varieties (D-121, Guard-53). How about HP1 and HP2?
Response: The average yield of HP1 (8709 kg/ha) was higher than 58 hybrids and also higher than one check variety (Guard-53, 8395 kg/ha), but less than the other check variety (D-121, 9197 kg/ha). On the other hand, the yield of HP2 (8833 kg/ha) was higher than 66 hybrids and also higher than one check hybrid (Guard-53), but less than other check hybrid (D-121). It was also observed that, over all the average yield of HP1and HP2 was higher in average of 104 hybrids in the four locations (Rice Research Institute Kala Shah Kaku (RRI KSK), Pakistan Agricultural Research Council Kala Shah Kaku (PARC, KSK), Emkay Shiehupura, Chaudhry Khair Din, Dera Ghazi Khan). Among seven locations both HP1 and HP2 performed very well (yield higher than the average grand mean of all locations yield 8608 kg/ha) in NUYT test 2020 (Supplementary table 1).
- L40: HP1 yield (7863 kg/ha) was higher than the average over-check varieties/hybrids. How about HP2 and HP3?
Response: Similarly in the year 2021 the test locations were increased the average yield of HP2 (7288 kg/ha) was higher than 25 hybrids and HP3 (7387 kg/ha) was higher than 42 hybrids. It was observed that, the yield of HP2 less than both the check hybrids (D-121(7518 kg/ha), Guard-53 (7341 kg/ha) and average yield of HP3 less than (D-121) and higher than (Guard-53). Over all HP2 and HP3 performed very well higher in average yield of 137 hybrids in the two locations (Rice Research Institute Kala Shah Kaku (RRI KSK), Pakistan Agricultural Research Council Kala Shah Kaku (PARC, KSK) IN NUYT test 2021(Supplementary table 2).
- The text in the introduction is too long with unnecessary sentences. I recommend that the authors should shorten it with the relevant sentences and paragraphs to this work.
Response: As per instructions the unnecessary sentences and paragraphs have been removed from the introduction part (Line 116-120, Line 132-138).
- The main objective of the study was to screening, genetic diversity, adaptation and high yield performance of honglian type hybrid rice in Pakistan. Please rephrase this sentence.
Response: As per instruction/suggestion the objective of the study has been rephrased as “The main objective of the study was to screening of Honglian type of hybrid rice on the basis of adaptability, genetic diversity and high yield performance in different locations of Pakistan (Line 159-161).
- “A total of 104 hybrids in 2020 and 137 hybrids in 2021 (including Honglian hybrid rice and check varieties) were tested under different ecological zones of Pakistan”. This paragraph is repeated in the L178-179.
Response: As per instruction the repeated sentence/paragraph has been removed from the main manuscript from the materials and methods part (Line 178-179).
- L268: Statistical significance is determination of relationships between two or more variables at level of significance of 1% and 5%. Rewrite this sentence.
Response: As per instruction the sentence has been revised as “Statistical significance is determination of relationships between two or more variables for the prevalence of significant variance for all studied traits that implicates the usefulness of the rice hybrids for genetics analysis at level of significance of 1% and 5%” (Line 268-271).
- Where is the data for section 3.2?
Response: Dear reviewer, the detail data of molecular markers (48) has been mentioned in the separate files as a supplementary table 3.
- What does figure 6 shown?
Response: Thanks for the comment. This figure showed the utilization frame work for increasing the resistance in crop varieties for the development and improvement of new plant populations with desirable traits through hybridization/recombination and back cross breeding method.
- I recommend that the authors should carefully check the manuscript. Present their work clearly and concisely. Select the tables and figures which are very important and give values to their text.
Response: Dear reviewer thank you very much for your nice comments. As per suggestion/instruction the manuscript has been improved. The table No. 7 and 8 has been removed from the manuscript (Line 356,365, 402-403). In results part line 361-365 has been removed regarding table 7 to 8. In discussion part line 523-532 has been removed regarding table 7 to 8. In discussion part unnecessary lines 580-588 have been removed. Remaining tables and figures values have been mentioned in text file according to instruction/suggestion.

Reviewer 3 Report (Previous Reviewer 3)
The authors have made great efforts to revise and edit the 2nd version of the manuscript. The paper is mostly fine now. In my opinion, the paper still needs a minor revision including
1. Carefully check all spellings, typing error, format following the requirements of the submitted journals. for example, line 563 subsection 4.4 check the spelling "Hnglian"
2. Recheck some sentences duplicated in introduction and discussion
3. Conclusion should be rearranged, for instance, narrate the significant findings of this study, then describe in short perspectives of this study
Author Response
Reviewer#3
Dear Reviewer and Editor of Agriculture,
Thank you very much for considering our manuscript entitled "Adaptation and High yield performance of Honglian type hybrid rice in Pakistan with desirable agricultural traits" (Manuscript ID: agriculture-2037805). We have carefully revised the manuscript according to the reviewers’ comments and editorial evaluation.
The authors have made great efforts to revise and edit the 2nd version of the manuscript. The paper is mostly fine now. In my opinion, the paper still needs a minor revision including
Response: Dear Reviewer, thanks for your very critical, nice review and good comments to make this manuscript better. The corrections are highlighted in red color
- Carefully check all spellings, typing error, format following the requirements of the submitted journals. for example, line 563 subsection 4.4 check the spelling "Hnglian"
Response: Thank you very much for pointing out.T he corrections have been made regarding spellings, typing error and format following the requirements of the journal in the main file as per suggestion. The spelling has been corrected from"Hnglian" to Honglian (Line 563).
- Recheck some sentences duplicated in introduction and discussion
Response: The sentence duplications have been corrected /removed from the introduction and discussion part as per suggestion. Unnecessary lines 116-120 have been removed and lines 132-138 the duplication part has been removed from introduction part. Line 577-587 has been removed from discussion part as per instruction. In discussion part unnecessary /duplicated lines 580-588 have been removed.
- Conclusion should be rearranged, for instance, narrate the significant findings of this study, then describe in short perspectives of this study
Response: The information has been rearranged in the conclusion part as per suggestion/instruction Line 633-654

This manuscript is a resubmission of an earlier submission. The following is a list of the peer review reports and author responses from that submission.
Round 1
Reviewer 1 Report
Dear Authors
01- This manuscript is poorly written, not meeting the minimum standards of a scientific paper. There are more personal feelings in presenting and analysing the results. This should be avoided.
This seems to be a test report of 3 hybrid rice strains HP1, HP2 and HP3 to the "National Uniform Yield Trials" and "Distinctness, Uniformity and Stability" according to the requirements of Pakistan Government; the author then upgraded and interpreted these three hybrid rice strains as ecologically suitable in Pakistan and even claimed to be "Moreover, it was considerably better, suggesting that Honglian-type hybrid rice varieties of Pakistan is less risky under climate change and global warming conditions"
This manuscript seemed numerous points that need correction and revision for clear understanding.
02- The abstract is too long. Please rewrite. Some formats and detail of the manuscript may not be MDPI format. The title is not related to the detail of the results. Please clearly specify the highlight of this study.
03- The introduction. No introduction of hybrid rice varieties in this study, their genetic and agronomic characteristics, or their origin, ...
What is the objective of this study? The introduction should be included 1) What is known 2) what is unknown / problem / Research GAP and 3) Hypothesis / how to solve this problem or close GAP.
Please more introduction about the Gap between the previous study and the importance of this study [the content in the discussion should be filtered and moved to the introduction part].
Please add more information in the introduction part. And indicate the objective of this study at the end of the introduction part.
04- The Method and material part: Too bad, too superficial and not enough information to repeat the experiments or verify.
- No one knows the profiles of hybrid rice strains (HP1, HP2, HP3, HP4, HP5, HP6, HLR-006, 105 WR-1906, Guard53, D-121 and KSK-13) used for this study. Please provide their full profiles such as their genetic, agronomic characteristics, and origin?
- Experimental location? In-house or in-field? Scale (m2 or ha?). Time of study, detailed planting start time and harvest time.
- Lines 137 - 139 page 4 "A total of 104 hybrids in 2020 and 137 hybrids in 2021 (including Honglian hybrid rice and check varieties) were tested under different ecological zones of Pakistan", please detail the 104 and 137 hybrid strains that were used in this study? Meanwhile, please specify the ecological parameters of these eco-zones; no one knows what are the ecological niches in this study. Otherwise, no one knows why (or how) the difference in ecological zones in Pakistan affected the yield performance of the experimental hybrid rice strains.
05- The Results and Discussion: can't read. The structure, writing style, comparative presentation and argumentation of the author are insufficient.
- Table 1 does not have information on the location, size and duration of the trial
- Table 2 is not the result of this study, these primers are collected from one or more different sources. The author does not understand that it should be in the Method Materials (or appendix) but why is the result part?
- Page 6, lines 196-197, "RRI, KSK; RRI, Dokri; PARC, KSK, Guard; Golarchi, Four Brothers Multan, Emaky Sheikhupura, CKD, DG Khan" what are they? The readers can not understand.
- The same for page 7, line 211
Almost results show only results (that cannot see in the figure or table), please explain the meaning of the results for clearly understood. Please try to design the results to meet the objective or related to the explanation in the text.
When discussed and compared with other research, how can compare directly? Please compare the meaning of the results, do not compare the results.
Almost results show only results (that cannot see in the figure or table), please explain the meaning of the results for clearly understood. Please try to design the results to meet the objective or related to the explanation in the text.
When discussed and compared with other research, how can compare directly? Please compare the meaning of the results, do not compare the results.
In the discussion part, please truly discuss, do not review or introduce again. (I recommend the author filter the details in the discussion part and then move to the introduction part).
06-The conclusions are so strange due to the objective of this study is not clear. The author's conclusion rambled, "It was also observed that under higher temperatures, the performance of Honglian type hybrid rice was stable and considerably better, which suggests that the cultivation of Honglian type hybrid rice varieties in Pakistan is less risky under the scenario of climate change and global warming due to high fertility rate. Such genetic material could be very useful as genetic resources for developing new elite breeding lines in changing ecosystems. The study will be equally beneficial to the farmers and the scientific community to strengthen the country's economy"
But none of the results presented in the manuscript support this claim.
Best wishes.
Reviewer 2 Report
Refer to the attached file.

Reviewer 3 Report
In this study, the authors have made great efforts to evaluate three hybrids (H1, H2 and H3) belonging to Honglian types via the different locations in Pakistan and also use SSR markers to study their genetic diversity. Various data were obtained and presented. However, the current paper was not well documented and organized; some parts were confusing and lack of information. Hence it needs to be critically revised and edited. The authors can find some useful comments and suggestions as below:
1. Reword the whole manuscript following the specific comments directly tracked in the pdf file
2. Must be presented the specific objective of this study at the end of the Introduction part
3. Totally revise the part of Materials and Methods (see comments in the pdf file). How did the author attain the data without the methods and protocols
4. There was no information related to the different locations where the hybrids were tested.
5. DNA analyses must be reworded and added more information regarding the primers, and should note the polymorphic markers in Fig 2.
6. The data obtained in NUYT and DUS trials must be clarified, which were done by the authors or DUS centers, hence cited references need.
8. Discussion should be reworded, especially subsection 4.3 must be rewritten (see comments in file). It looks like the project proposal, not a scientific paper.
9. The conclusion must be reworded and should focus on the significant results obtained in this study (see comments in file)
